# Storage Efficient and Dynamic Flexible Runtime Channel Pruning via Deep Reinforcement Learning

**Jianda Chen**
Nanyang Technological University, Singapore
jianda001@e.ntu.edu.sg

**Shangyu Chen**
Nanyang Technological University, Singapore
schen025@e.ntu.edu.sg

**Sinno Jialin Pan**
Nanyang Technological University, Singapore
sinnopan@ntu.edu.sg

## Abstract

In this paper, we propose a deep reinforcement learning (DRL) based framework to efficiently perform runtime channel pruning on convolutional neural networks (CNNs). Our DRL-based framework aims to learn a pruning strategy to determine how many and which channels to be pruned in each convolutional layer, depending on each individual input instance at runtime. Unlike existing runtime pruning methods which require to store all channels parameters for inference, our framework can reduce parameters storage consumption by introducing a static pruning component. Comparison experimental results with existing runtime and static pruning methods on state-of-the-art CNNs demonstrate that our proposed framework is able to provide a tradeoff between dynamic flexibility and storage efficiency in runtime channel pruning.

## 1 Introduction

In recent years, convolutional neural networks (CNNs) have demonstrated remarkable performance in various computer vision tasks [17, 30, 8, 7, 36, 18, 7, 37, 4]. However, since most state-of-the-art CNNs require expensive computation power for inference and huge storage space to store large amount of parameters, the limitation of energy, computation and storage on mobile or edge devices has become the major bottleneck on real-world deployments of CNNs. Existing studies have been focused on speeding up the execution of CNNs for inference on edge devices by model compression using matrix decomposition [3, 25], network quantization [2], network pruning [5], etc. Among these approaches, channel pruning, which discards an entire input or output channel and keeps the rest of the model with structures, has shown promising performance [11, 24, 38, 26].

Most channel pruning approaches can be categorized into two types: runtime and static. Static approaches aim to evaluate the importance of each channel over the whole training dataset and remove the least important channels to minimize the loss of performance after pruning. By permanently pruning a number of channels, the computation and storage cost of a CNN can be dramatically reduced when being deployed, and the inference execution can be accelerated consequently. On the other hand, runtime approaches have been recently proposed to achieve dynamic channel pruning on each individual instance [6]. To be specific, the goal of runtime approaches aims to evaluate the channel importance at runtime, which is assumed to be different on different input instances. By pruning channels dynamically, different pruned structures can be considered as different routing of data stream inside CNNs. This kind of approaches is able to significantly improve the representation capability of a CNN, and thus achieve better performance in terms of prediction accuracy compared with static approaches. However, previous runtime approaches trade storage cost off dynamic flexibility. To

achieve dynamic pruning on different individual instances, all parameters of kernels are required to be stored (or even more parameters are introduced). This makes runtime approaches not applicable on resource-limited edge devices. Moreover, most of previous runtime approaches only evaluate the importance among channels in each single layer independently, without considering the difference in efficiency among layers.

In this paper, to address the aforementioned issues of runtime channel pruning approaches, we propose a deep reinforcement learning (DRL) based pruning framework. Basically, we aim to apply DRL to prune CNNs by maximizing received rewards, which are designed to satisfy the overall budget constraints along side with the network's training accuracy. Note that automatic channel pruning by DRL is a challenging task because the action space is usually very huge. Specifically, the discrete action space for the DRL agent is as large as the number of channels at each layer, and the action space may vary among layers since there are different numbers of channels in different layers. To facilitate pruning CNNs by DRL, for each layer, we first design a novel prediction component to estimate the importance of channels, and then develop a DRL-based component to learn the sparsity ratio of the layer, i.e., how many channels should be pruned.

Different from previous runtime channel pruning approaches, which only learn runtime importance of each channel, we propose to learn both runtime importance and additionally static importance for each channel. While runtime importance maintains the saliency of specific channels for each individual input, the static importance captures the overall saliency of the corresponding channel among the whole dataset. According to each type of channel importance, we further design different DRL agents (i.e., a runtime agent and a static agent) to learn a sparsity ratio in a layer-wise manner. The sparsity ratio learned by the runtime agent together with the estimated runtime importance of channels is used to generate runtime pruning structures, while the sparsity ratio learned by the static agent together with the estimated static importance of channels is used to generate static (permanent) pruning structures. By considering both the pruning structures, our framework is able to provide a trade-off between storage efficiency and dynamic flexibility for runtime channel pruning.

In summary, our contributions are 2-fold. First, we propose to prune channels by taking both runtime and static information of the environment into consideration. Runtime information endows pruning with flexibility based on different input instances while static information reduces the number of parameters in deployment, leading to storage reduction, which cannot be achieved by conventional runtime pruning approaches. Second, we propose to use DRL to determine sparsity ratios, which is different from the previous pruning approaches that manually set sparsity ratios. The codes of our method can be found at `https://github.com/jianda-chen/static_dynamic_rl_pruning`.

## 2 Related Work and Preliminary

**Structure Pruning**    [33] pioneered structure pruning in deep neural network by imposing the $L_{2,1}$ norm in training. Under the same framework, [22] regarded parameters in batch normalization as channel selection signal, which is minimized to achieve pruning during training. [11] formulated channel pruning into a two-step iterative process including LASSO regression based channel selection and least square reconstruction. [24] formulated channel pruning as minimization of difference of output features, which is solved by greedy selection. [38] further considered early prediction, reconstruction loss and final loss to select importance channels. [23] proposed to use layer input to learn channel importance, which is then binarized for pruning. Overall, structure pruning methods accelerate inference by producing regular and compact model. However, this brought regularness requires preserving more parameters to ensure performance.

**Dynamic Pruning**    Dynamic pruning provides different pruning strategies according to input data. [32] proposed to reduce computation by skipping layers or channels based on the analysis of input features. [6] applied the same framework while extended features selection in both input and output features. Similarly, [20] introduced multiple branches for runtime inference according to inputs. A gating module is learnt to guide the flow of feature maps. [1] learned to choose the components of a deep network to be evaluated for each input adaptively. Early exit is introduced to accelerate computation. Dynamic pruning adaptively takes different actions for different inputs, which is able to accelerate the overall inference time. However, the original high-precision model needs to be stored, together with extra parameters for making specified pruning actions. [27] proposed to learn routers to route layers output to different next layers, in order to adjust a network to multi-task learning.

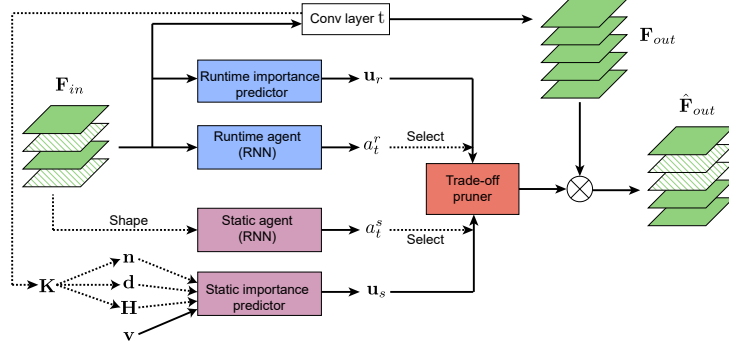

Figure 1: Our proposed DRL-based runtime pruning framework. Solid arrows indicate differentiable dataflow. Dash arrows indicate dataflow that is non-differentiable or does not require differentiation.

**DRL for Pruning**    Channel selection is on trial using DRL. [19] trained a LSTM model to remember and provide channel pruning strategy for backbone CNN model, which is conducted using reinforcement learning techniques. [10] proposed to determine the compression ratio in each layer by training an agent regarding the pruning-retraining process as an environment.

**Reinforcement Learning**    We consider a standard setup of reinforcement learning: an agent sequentially takes actions over a sequence of time steps in an environment, in order to maximize the cumulative reward [31]. This problem can be formulated as a Markov Decision Process (MDP) of a tuple $(\mathcal{S}, \mathcal{A}, \mathcal{P}, R, \gamma)$, where $\mathcal{S}$ is the state space, $\mathcal{A}$ is the action space, $\mathcal{P} : \mathcal{S} \times \mathcal{A} \times \mathcal{S} \rightarrow [0, 1]$ is transition probabilities, $R : \mathcal{S} \times \mathcal{A} \rightarrow \mathbb{R}$ is the reward function, and $\gamma \in [0, 1)$ is the discount factor. The goal of reinforcement learning is to learn a policy $\pi(a|s)$ that maximizes the objective of discounted cumulative rewards over finite time steps as $\max_\pi \sum_{t=0}^{T} \gamma^t R(s_t, a_t)$, where $s_t \in \mathcal{S}$ and $a_t \in \mathcal{A}$ are state and taken action at time step $t$ respectively.

## 3    DRL-based Runtime Pruning Framework

The overview of our proposed framework is presented in Fig. 1, which shows our pruning procedure in the $t$-th convolutional layer. To prune convolutional layer $t$, we 1) learn two types of *channel importance*: *static* channel importance $\mathbf{u}_s$ and *runtime* channel importance $\mathbf{u}_r$; 2) learn two DRL agents, the *runtime agent* and the *static agent*, producing action $a_t^r$ and $a_t^s$ which are defined as sparsity ratios of runtime and static pruning, respectively; 3) perform a *trade-off pruner* to balance the runtime and static pruning results.

Both runtime channel importance $\mathbf{u}_r \in \mathbb{R}^C$ and static channel importance $\mathbf{u}_s \in \mathbb{R}^C$ indicate the channel importance of the full precision output feature map $\mathbf{F}_{out}$ of a convolution layer, where $C$ is the number of channels in $\mathbf{F}_{out}$. Channels are selected to be pruned according to the values of each element in $\mathbf{u}_r$ and $\mathbf{u}_s$, and how many channels to be selected is decided by the sparsity ratios $a_t^r$ and $a_t^s$, respectively. The balanced pruning result is represented by two items, a decision mask $\mathbf{m}$ of binary values (1/0) to indicates which channels to be pruned (1: pruned, 0: preserved) and a unified channel importance vector $\mathbf{u}$. $\mathbf{m}$ and $\mathbf{u}$ are generated by trade-off pruner $g$ as $[\mathbf{m}, \mathbf{u}] = g(\mathbf{u}_r, \mathbf{u}_s, a_t^r, a_t^s)$, where $a_t^r, a_t^s \in \mathbb{R}$, $\mathbf{u} \in \mathbb{R}^C$ and $\mathbf{m} \in \{0, 1\}^C$. The final output after pruning is constructed by multiplying the full precision output feature map $\mathbf{F}_{out}$, by $\mathbf{1} - \mathbf{m}$ and $\mathbf{u}$ as,

$$\hat{\mathbf{F}}_{out} = \mathbf{F}_{out} \otimes (\mathbf{1} - \mathbf{m}) \otimes \mathbf{u}, \tag{1}$$

where $\otimes$ is the broadcast element-wise multiplier, and $\mathbf{1}$ is the matrix of the same size as $\mathbf{m}$ with all the elements being 1. In the following, we introduce how to learn the runtime channel importance vector $\mathbf{u}_r$ and the static channel importance vector $\mathbf{u}_s$ in Sec. 3.1, how to construct the trade-off pruner $g(\cdot)$ in Sec. 3.2, and how to design the two DRL agents to predict $a_t^r$ and $a_t^s$ in Sec. 3.3.

## 3.1 Learnable Channel Importance

We consider that a convolutional layer takes input of feature map $\mathbf{F}_{in} \in \mathbb{R}^{C_{in} \times H_{in} \times W_{in}}$ and generates an output feature map $\mathbf{F}_{out} \in \mathbb{R}^{C_{out} \times H_{out} \times W_{out}}$, where $C_*$, $H_*$ and $W_*$ are the number of channels, width and height of the feature map $\mathbf{F}_*$, respectively. Each element of the channel importance vectors $\mathbf{u}_r \in \mathbb{R}^{C_{out}}$ and $\mathbf{u}_s \in \mathbb{R}^{C_{out}}$ represents the importance value of the corresponding channel, respectively. In the following, we drop the subscript $_{out}$ for simplicity in presentation.

**Runtime Channel Importance**  The runtime channel importance $\mathbf{u}_r$ of output feature $\mathbf{F}_{out}$ is predicted by a *runtime importance predictor* $f(\cdot)$, which takes $\mathbf{F}_{in}$ as input. Therefore, $\mathbf{u}_r$ can be considered as a function of $\mathbf{F}_{in}$, whose values vary over different input instances. In this paper, we design a subnetwork to approximate $f(\cdot)$, which is expected to be small and computationally efficient. Similar to many existing dynamic pruning methods [6, 13], we use global pooling as the first layer in $f(\cdot)$, because global pooling is computationally efficient and it can reduce the dimension of $\mathbf{F}_{in}$ dramatically. We then feed the output of global pooling into a fully-connected layer without any activation function. The output of fully-connected layer is the runtime channel importance vector $\mathbf{u}_r$.

**Static Channel Importance**  The static channel importance $\mathbf{u}_s$ is to capture the global information for pruning, and thus is learned from the whole dataset. An *static importance predictor* $h(\cdot)$ is introduced to predict $\mathbf{u}_s$ by taking convolutional filters' features and a learnable vector $\mathbf{v} \in \mathbb{R}^{C_{out}}$ as input. We denotes $\mathbf{K} \in \mathbb{R}^{C_{out} \times C_{in} \times k \times k}$ as the convolutional filters in layer $t$, where $k$ is kernel size. Three types of features are designed upon convolutional filter $\mathbf{K}$: norm $\mathbf{n} \in \mathbb{R}^{C_{out}}$, filter distance $\mathbf{d} \in \mathbb{R}^{C_{out}}$ and filter Hessian diagonal $\mathbf{H} \in \mathbb{R}^{C_{out}}$. The norm value $\mathbf{n}^{(i)}$ for $i$-th output channel is defined as Frobenius norm of $i$-th filter $\mathbf{K}^{(i)}$: $\mathbf{n}^{(i)} = \left\| \mathbf{K}^{(i)} \right\|_F$, where $\| \cdot \|_F$ stands for Frobenius norm. Inspired by FPGM [9], filter distance is an import feature for pruning. We define distance $\mathbf{d}^{(i)}$ as the Euclidean distance between the $i$-th filter $\mathbf{K}^{(i)}$ and a "mean" filter: $\mathbf{d}^{(i)} = \left\| \mathbf{K}^{(i)} - \frac{1}{C_{out}} \sum_{i=1}^{C_{out}} \mathbf{K}^{(i)} \right\|_F$. Due to computation complexity of Hessian matrix, we approximate Hessian diagonal using square of first-order Taylor expansion. Therefore our filter Hessian diagonal $\mathbf{H}^{(i)}$ for $i$-th filter is defined as the sum of approximated Hessian diagonal of all weights in $i$-th filter $\mathbf{K}^{(i)}$: $\mathbf{H}^{(i)} = \left\| \frac{\partial \mathcal{L}_{CNN}}{\partial \mathbf{K}^{(i)}} \otimes \mathbf{K}^{(i)} \right\|_F^2$, where $\otimes$ is element-wise multiplication and $\mathcal{L}_{CNN}$ is the CNN loss. The first-order gradient $\frac{\partial \mathcal{L}_{CNN}}{\partial \mathbf{K}^{(i)}}$ is evaluated on the whole training dataset.

The static channel importance $\mathbf{u}_s$ is then generated by the static importance predictor $h(\cdot)$ taking input of $\mathbf{n}$, $\mathbf{d}$, $\mathbf{H}$ and $\mathbf{v}$ as $\mathbf{u}_s^{(i)} = h(\mathbf{n}^{(i)}, \mathbf{d}^{(i)}, \mathbf{H}^{(i)}, \mathbf{v}^{(i)})$, where $\mathbf{u}_s^{(i)}$ represents the static importance value of $i$-th output channel and $\mathbf{u}_s$ is concatenation of $\mathbf{u}_s^{(i)}$. $h(\cdot)$ is approximated by a subnetwork which consists of two fully-connected layers with one SeLU [15] activation layer in the middle. The reason of not choosing ReLU is "dead ReLU", which indicates it may always output zeros and stops backpropagating gradient. The three feature vectors $\mathbf{n}$, $\mathbf{d}$ and $\mathbf{H}$ are generated by pretrained backbone CNN as introduced above and fixed during further training. Meanwhile $\mathbf{v}$ is randomly initialized and learned through backpropagation. The subnetwork $h(\cdot)$ shares parameters among all convolutional layers in order to learn importance globally, while $\mathbf{v}$ is layer specific to learn locally.

## 3.2 The Trade-off Pruner

We propose a *trade-off pruner* to generate a unified channel pruning decision. The goal of the trade-off pruner is to 1) prune those channels that are decided to be pruned by both of the runtime and the static pruning components, and 2) prune a portion of the rest channels by weighted votes from the two components.

Which channels to be preserved / pruned at runtime are determined according to the values of runtime channel importance $\mathbf{u}_r$. Let $\mathbf{m}_r \in \{0, 1\}^C$ denotes a decision mask for pruning, where if the value is 0, then the corresponding channel is preserved, otherwise pruned. For now, suppose a sparsity ratio $a_t^r$ for runtime pruning has already been generated via the dynamic DRL agent, which will be introduced in Sec. 3.3. We then prune $(C - \lceil a_t^r C \rceil)$ channels with the smallest importance values in $\mathbf{u}_r$. Accordingly, the value of an element in $\mathbf{m}_r$ is set to be 1 if the corresponding channel is pruned, otherwise 0. Similarly, for static pruning, given static importance $\mathbf{u}_s$ and a sparsity ratio $a_t^s$ learned by the static DRL agent, $(C - \lceil a_t^s C \rceil)$ channels with smallest importance values in $\mathbf{u}_s$ are pruned, and a mask $\mathbf{m}_s \in \{0, 1\}^C$ is generated to indicate the static pruning results.

With the runtime and the static pruning decisions, $\mathbf{m}_r$ and $\mathbf{m}_s$, our trade-off pruner generates a unified pruning result. To be specific, we define the mask representing channels pruned by both decisions as $\mathbf{m}_o = \mathbf{m}_s \wedge \mathbf{m}_r$, where $\wedge$ is element-wise logical AND and 1/0 in mask represents logical *true* or *false*. The channels indicated to be pruned by $\mathbf{m}_o$ (i.e., the corresponding values are 1) are pruned in final. The channels which are determined to be pruned by $\mathbf{m}_r$ but not by $\mathbf{m}_s$ can be represented by a new mask $\overline{\mathbf{m}}_r = \mathbf{m}_r - \mathbf{m}_o$. Similarly, the channels which are determined to be pruned by $\mathbf{m}_r$ but not by $\mathbf{m}_s$ can be represented by another new mask $\overline{\mathbf{m}}_s = \mathbf{m}_s - \mathbf{m}_o$.

To control the trade-off between $\mathbf{m}_r$ and $\mathbf{m}_s$, we define a rate $R_r$ denoting how much we trust the pruning decision made by $\mathbf{m}_r$, while $1 - R_r$ is for $\mathbf{m}_s$. That means the channels selected by $\overline{\mathbf{m}}_r$ will be finally pruned with the rate $R_r$. Specifically, the number of channels which are selected by $\overline{\mathbf{m}}_r$ and finally will be pruned is $C_r^{'} = \lfloor R_r(\mathbf{1}^\top \overline{\mathbf{m}}_r) \rfloor$, where $\mathbf{1}^\top \overline{\mathbf{m}}_r$ returns the number of channels selected by $\overline{\mathbf{m}}_r$. We then select the first $C_r^{'}$-smallest important channels which are recommended to be pruned by $\overline{\mathbf{m}}_r$ to form a mask $\hat{\mathbf{m}}_r$. Similarly, for static pruning, we select the first $C_s^{'}$-smallest important channels which are recommended to be pruned by $\overline{\mathbf{m}}_s$ to form another mask $\hat{\mathbf{m}}_s$, where $C_s^{'} = \lfloor (1 - R_r)(\mathbf{1}^\top \overline{\mathbf{m}}_s) \rfloor$.

The final trade-off pruning mask is defined as $\mathbf{m} = \mathbf{m}_o + \hat{\mathbf{m}}_r + \hat{\mathbf{m}}_s$, and the unified channel importance is simply defined as $\mathbf{u} = \mathbf{u}_r \otimes \mathbf{u}_s$. With the trade-off pruning mask $\mathbf{m}$ and the unified channel importance $\mathbf{u}$, the pruned output feature $\hat{\mathbf{F}}_{out}$ can be generated by Eq. 1.

### 3.3 DRL-based Sparsity Ratio Estimation

In this section, we present how to formulate the problems of learning the ratios $a_t^s$ and $a_t^r$ for static pruning and runtime pruning, as a MDP, and solve it via DRL, respectively.

**The Runtime DRL Agent**    In the MDP for runtime pruning, we consider the $t$-th layer of the network as the $t$-th timestamp. The details of the MDP are listed as follows.

**State**    Given an input feature map $\mathbf{F}_{in}$ of layer $t$, we pass it to a global pooling layer to reduce its dimension to $\mathbb{R}^{C_{in}}$, where $C_{in}$ is the number of input channel of layer $t$. Since $C_{in}$ varies among layers, we feed the output of global pooling to a layer-dependent encoder to project it to a fix-length vector $\mathbf{s}_t^r$ with dimensions of 128, which is considered as as a *state* representation of DRL in the context of runtime pruning.

**Action**    The *action* $a_t^r$ is defined as the sparsity ratio at layer $t$. Existing DRL-base pruning method RNP [19] uses a unified discrete actions space with $k$ actions which are too coarse to achieve high accuracy. However, fine-grained discrete action space as large as number of channels suffers from exploration difficulty. Therefore, instead of using discrete action spaces, we propose a continuous action space with action $a_t^r \in (0, 1]$. To avoid over-pruning the filters and crashing in training, we set a minimum sparsity ratio $+\alpha$ such that $a_t^r \in (+\alpha, 1]$.

**Reward**    The reward function is proposed to consider both network accuracy and computation budget. We define the accuracy relative reward based on the loss of pruned backbone network as $R_{acc}^r = -\mathcal{L}_{CNN}$, where $\mathcal{L}_{CNN}$ is the CNN loss, and it may vary in scale among different training stage, i.e. large at beginning of training and small near convergence. To avoid the instability brought by the reward scale, $R_{acc}^r$ is normalized by a moving average via $R_{acc}^{r'} = R_{acc}^r/\beta_b$, where $\beta_b = \lambda\beta_{b-1} + (1 - \lambda)R_{acc}^r$ is the moving average at the $b$-th training batch and $\lambda$ is the weight.

To force computation of the pruned network under a given computation budget, we define a exponential reward function of budget regarding reward $R_{bud}^r = 1 - \exp(\alpha_1(B_{com} - \overline{B}_{com}))$ if $B_{com} > \overline{B}_{com}$, otherwise, $R_{bud}^r = 0$, where $B_{com}$ is the computation consumption, which is calculated based on the current of pruned strategy, and $\overline{B}_{com}$ is the given computation budget constraint. Finally we sum up the two rewards to form sparse rewards $R_t^r = R_{acc}^{r'} + R_{bud}^r$ if $t = T$, and $R_t^r = 0$ if $t < T$.

**Actor-Critic Agent**    To solve the continuous action space problem, we choose a commonly used actor-critic agent with a Gaussian policy. Actor-critic agent consists of two components: 1) actor outputs the mean and variance to form a Gaussian policy where the continuous action is sampled from; 2) critic outputs a scalar predicting the future discounted accumulated reward and assists the

policy training. Actor network and Critic network share one-layer RNN with hidden state size of 128 which takes state $\mathbf{s}_t^r$ as input. The output of RNN is fed into actor specific network constructed by two branches of fully-connected layers, leading to the mean and variance of the Gaussian policy. The action is sampled for the Gaussian distribution outputed by the actor: $a_t^r \sim \mathcal{N}(\mu(\mathbf{s}_t^r; \theta^r), \sigma(\mathbf{s}_t^r; \theta^r))$, where $\mu(\mathbf{s}_t^r; \theta^r)$ and $\sigma(\mathbf{s}_t^r; \theta^r)$ is the mean and variance outputed from actor network. The Critic specific network has one fully-connected layer after the shared RNN, and outputs the predictive value $V(\mathbf{s}_t^r; \theta^r)$.

To optimize the actor-critic agent, Proximal Policy Optimization (PPO) [29] is used. Note that we relax the action $a_t^r$ to $(-\infty, +\infty)$ in PPO, and use truncate function to clip $a_t^r$ in $(+\alpha, 1]$ when perform pruning. Besides, an additional regularizer $\mathcal{L}_a$ is introduced to restrict the relaxed $a_t^r$ staying in range $(+\alpha, 1]$ as $\mathcal{L}_a = \frac{1}{2}||a_t^r - \max(\min(a_t^r, 1), +\alpha)||_2^2$.

**The Static DRL Agent**   Similar to runtime pruning, the MDP in static pruning is also formulated layer-by-layer. The difference against runtime pruning is the definition of state and reward. The state $\mathbf{s}_t^s$ in static pruning is defined as the full shape of $\mathbf{F}_{out}$, and does not depend on $\mathbf{F}_{out}$ and the current input data. **Action** $a_t^s$ is sampled from actor's outputed Gaussian policy, and it represents the sparsity ratio at layer $t$. The **reward** function takes both network accuracy and parameters budget into consideration. The accuracy relative is defined as the same as that in runtime pruning, i.e., $R_{acc}^s = R_{acc}^{r'}$. To reduce the number of parameters of network to satisfy the parameters storage budget, the parameters relative reward is defined in an exponential form as $R_{param}^s = 1 - \exp(\alpha_2(B_{param} - \overline{B}_{param}))$ if $B_{param} > \overline{B}_{param}$, otherwise $R_{param}^s = 0$, where $B_{param}$ is the number of preserved parameters after static pruning and $\overline{B}_{param}$ is the parameters storage budget.

**Actor-Critic Agent**   This agent is similar to the one in runtime pruning. It has the same architecture as runtime pruning but differs in introducing a fully-connected layer as the encoder before RNN. This agent is also optimized by PPO.

## 3.4   Training and Inference

Alg. 1 illustrates the training process of our method. Given a pretrained backbone CNN, firstly we finetune the CNN and train runtime importance predictor jointly (line 5), with sparsity $a_t^r = 1$ and fixed all static pruning importance $\mathbf{u}_s$ to 1. We then remove the restriction on the static pruning importance $\mathbf{u}_s$ which is now generated by importance predictor $h(\cdot)$ and layer-specific learnable vector $\mathbf{v}$. We train static pruning $h(\cdot)$, $\mathbf{v}$, the backbone CNN and the runtime importance predictor (line 8), with sparsity $a_t^s = 1$ and runtime pruning sparsity fixed as $a_t^r = 1$. After finetuning, we use the DRL agents to predict the sparsity given computation and storage constraints. The DRL agents and the CNN with runtime/static importance are trained in alternating manner: We first fix the CNN as well as runtime/static importance and train two DRL agents,

**Algorithm 1** Training process

---
1: **Input:**  pretrained backbone CNN, computation budget $\bar{B}_{com}$, storage budget $\bar{B}_{param}$
2: **Output:** CNN, importance predictor $f(\cdot)$, static importance $u_s$, runtime and static DRL agents
3: $\mathbf{u_s} \leftarrow \mathbf{1}, a_t^s \leftarrow 1, a_t^r \leftarrow 1$
4: **while** not converge **do**
5:     fix $\mathbf{u_s}$, update $f(\cdot)$ and CNN
6: **end while**
7: **while** not converge **do**
8:     Update $h(\cdot)$, $\mathbf{v}$, $f(\cdot)$ and CNN {$\mathbf{u_s}$ is generated by $h(\cdot)$ and $\mathbf{v}$}
9: **end while**
10: Use DRL agents to predict actions $a_t^r$ and $a_t^s$ at each layer $t$
11: **while** not converge **do**
12:     **for** $i \in 1...N_1$ **do**
13:         Compute rewards $R_t^r$ and $R_t^s$ using budget $\bar{B}_{com}$ and $\bar{B}_{param}$
14:         Fix $\mathbf{u_s}$, $f(\cdot)$ and CNN, update DRL agents
15:     **end for**
16:     **for** $i \in 1...N_2$ **do**
17:         Fix DRL agents, update $h(\cdot)$, $\mathbf{v}$, $f(\cdot)$ and CNN
18:     **end for**
19: **end while**

---

regarding the CNN as environments (line 12 to line 15). We then fix two agents and finetune the CNN and runtime/static importance (line 16 to line 18). We repeat these two steps until convergence.

For CIFAR-10, we train ($N_1 = 1560$) batches (4 epochs) for DRL agents (line 12 to line 15) and ($N_2 = 780$) batches (2 epochs) for $h(\cdot)$, $\mathbf{v}$, $f(\cdot)$ and CNN (line 16 to 18) at each iteration. The total number of iterations is 40 (for line 11 to 19). For ImageNet, at each iteration of training, $N_1 = 1200$ and $N_2 = 600$ except last iteration. The total number of iterations is 64. At last iteration, $N_2$

= 200,000 for finetuning the CNN. We use Adam optimizer for both DRL agent and CNN, and set learning rate to $10^{-6}$ for the DRL agents. For CNN finetuning and runtime/static importance training, the learning rate is $10^{-3}$ on CIFAR-10. On ImageNet ILSVRC2012, the learning rate is $10^{-4}$.

**Inference**   While runtime pruning performs for individual instance dynamically, static pruning strategy does not depend on individual input data points. Therefore, the action $a_t^s$ predicted by static agent and the static importance vector $\mathbf{u_s}$ generated by static importance predictor $h(\cdot)$ are fixed, regardless of input instances. We can consequently remove static agent and $h(\cdot)$ in inference, and eliminate the storage and computation cost of them. With the action $a_t^s$ and hyper-parameter $R_r$, we can decide which filters can be pruned permanently. Specifically, channels with $((1 - a_t^s)(1 - R_r))$-smallest static importance values are pruned permanently.

## 4   Experiment

We evaluate our DRL pruning framework on two benchmark datasets: CIFAR-10 [16] and ImageNet ILSVRC2012 [28]. Our goal is to verify our framework is able to achieve comparable or even better performance in terms of prediction accuracy as other state-of-the-art channel pruning methods, but save much more storage space. We further analyze the effect of hyper-parameters and different sparsity settings on CIFAR-10. For CIFAR-10, we use M-CifarNet [35] as the backbone CNN. On ImageNet ILSVRC2012, ResNet-18 [8] and MobileNet [12] are used as the backbone CNN.

**Experimental results on CIFAR-10**   We compare our proposed method with the following state-of-the-art runtime pruning methods: FBS [6], RNP [19] on CIFAR-10. The comparison results at sparsity 0.5 and 0.7 are shown in Table 1 and Table 2 respectively. The sparsity is defined as the ratio of preserved output channels after pruning at every layer.

| Method | Baseline acc. | Acc. | $\Delta acc.$ | Speed-up | #Params | GPU Time | CPU Time |
|---|---|---|---|---|---|---|---|
| FBS [6] | 91.37 | 89.88 | -1.49 | **3.93×** | 1.11× | 10.9 ms | 172.0ms |
| RNP [19] | 92.07 | 84.93 | -7.14 | 3.56× | 1.00× | 11.1 ms | 175.3ms |
| ours ($R_r = 1$) | 92.07 | 91.425 | **-0.645** | 3.92× | 1.31× | 11.2 ms | 178.3ms |
| ours ($R_r = 0.5$) | 92.07 | 91.228 | **-0.842** | 3.92× | **0.78×** | **9.8 ms** | **110.7ms** |

Table 1: Comparison with state-of-the-art runtime pruning methods on CIFAR-10 at sparsity 0.5. Speed-up is calculated on MACs.

| Method | Baseline acc. | Acc. | $\Delta acc.$ | Speed-up | #Params |
|---|---|---|---|---|---|
| FBS [6] | 91.37 | 91.23 | -0.14 | **2×** | 1.11× |
| ours ($R_r = 1.0$) | 92.07 | 93.184 | **1.114** | 1.99× | **1.31×** |
| ours ($R_r = 0.5$) | 92.07 | 92.714 | **0.629** | 1.99× | **0.97×** |

Table 2: Comparison with state-of-the-art runtime pruning methods on CIFAR-10 at sparsity 0.7. Speed-up is calculated on MACs.

Note that for a fair comparison with other methods, the computation and storage budget constraints in our method is calculated according to the sparsity of other methods. Under these constraints, our method does not necessarily lead to the same sparsity as other methods in each layer. RNP cannot set exact sparsity ratio. Instead, its average sparsity ratio is accessible only during testing, which is 0.537 in Table 1. The result of FBS is reproduced using the released code[1]. The column *#Params* represents the number of parameters compared to the backbone CNN. Speed-up is calculated on MACs. The column of GPU time is the time of forwarding a batch of 256 images on an NVIDIA P100 GPU and CPU time is on CPU.

Table 1 shows that our method outperforms other state-of-the-art methods, achieving highest accuracy at an overall sparsity ratio of 0.5. Our method has very close computation speed-up compared to FBS, but outperforms FBS around 0.65% to 0.86%. When the runtime pruning strategy is solely considered by setting $R_r = 1$, our method surpasses other comparison methods, indicating that our DRL-based framework improves the performance of channel runtime pruning. By balancing runtime and static

pruning via setting $R_r = 0.5$, our method reduces the number of the overall stored parameters and achieves lower accuracy drop than other methods. Table 2 shows that our method outperforms FBS at sparsity of 0.7. When $R_r = 0.5$, our method achieves better performance than the baseline CNN with $2\times$ speed-up and contains less parameters.

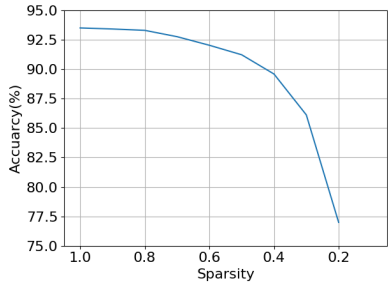

Figure 2: Accuracy drop for M-CifarNet on CIFAR-10 with computational budget.

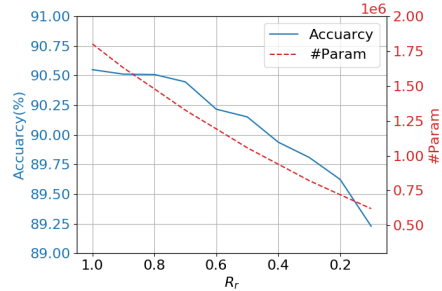

Figure 3: Trade-off between runtime and static pruning at sparsity 0.45.

**Trade-off between Runtime and Static Pruning**  Fig. 2 shows the performance of various sparsity ratios in our method. Again, our method does not prune with one single sparsity ratio for all layers, but uses the sparsity ratio to calculate computation and storage constraints, with which the sparsity ratio is learned for each layer. Fig. 2 demonstrates that our method holds the accuracy when sparsity is larger than about 0.5, which corresponds to about $4\times$ computational acceleration.

We also study the relation between $R_r$ and network compactness in our framework. Fig. 3 demonstrates the impact of $R_r$ when sparsity is 0.45. The hyper-parameter $R_r$ determines how much we trust about runtime pruning. With $R_r$ close to 1, the accuracy becomes higher due to the more dynamic network flexibility but the space of the parameters storage also increases. When $R_r$ diminishes, the network accuracy decreases but the parameter storage is reduced.

| Method | Baseline acc. | Acc. | $\Delta acc.$ |
|---|---|---|---|
| ours (runtime only $R_r = 1$) | 92.07 | 91.425 | **-0.645** |
| ours ($R_r = 0.5$) | 92.07 | 91.228 | **-0.842** |
| ours (separate static and runtime) | 92.07 | 90.03 | -2.04 |
| FPGM+FBS | 92.07 | 90.456 | -1.614 |

Table 3: Comparison to methods with separately static and runtime pruning on CIFAR-10 at sparsity 0.5.

**Comparison to Separately Static and Runtime Pruning**  In this section, we compare our method with two additional baseline methods. One is a variation of our method by separately training static pruning and runtime pruning. In this method, we start from a pretrained backbone CNN, $f(\cdot)$ and $u_s$. Then we add the static DRL agent to prune channels statically by learning the static policy and $u_s$. Finally, we add the runtime DRL agent to prune channels dynamically, by fixing the static DRL agent and $u_s$, and updating the runtime DRL agent and $f(\cdot)$ only. Another method is to combine state-of-the-art static and runtime pruning methods. We start from a pretrain backbone CNN, and then prune channels with the static pruning method FPGM [9], and finally prune channels with the runtime method FBS [6]. The experimental results are shown in Table 3.

**Hyper-parameter choosing: learning rate**  In this section, we set up an experiment with comparing various learning rates for training procedure of our proposed method. Table 4 shows the accuracy results of various learning rates settings. The first column indicates the learning rate for DRL agents and second column indicates the learning rate for training $f(\cdot)$ and CNN. It shows that our learning rates setting, $10^{-6}$ for DRL agents and $10^{-3}$ for $f(\cdot)$ and CNN, achieves the best accuracy performance.

| DRL agents lr | CNN & $f(\cdot)$ lr | Acc. |
|---|---|---|
| $10^{-6}$ | $10^{-3}$ | **91.23** |
| $10^{-6}$ | $10^{-2}$ | 10.00 |
| $10^{-6}$ | $10^{-4}$ | 90.67 |
| $10^{-5}$ | $10^{-3}$ | 89.95 |
| $10^{-7}$ | $10^{-3}$ | 90.79 |

Table 4: Comparison of various learning rates on CIFAR-10.

**Experimental results on ImageNet ILSVRC2012** We compare our method with state-of-the-art channel pruning methods on ImageNet ILSVRC2012. In the first experiment as shown in Table 5, we use ResNet-18 as the backbone CNN. Among the methods for comparison, FBS [6] and CGNN [14] are runtime pruning methods and AMC [10] is DRL-based static pruning. The overall sparsity ratio of our method is 0.7, which is under the same setting of FBS. Our method with $R_r = 0.5$ achieves the smallest top-1 accuracy drop compared with other methods, and also achieves the highest top-1 accuracy after pruning. Our method has very close MACs to FBS, while the number of preserved parameters is reduced to $81.2\%$ of the baseline.

We also implement the our method on MobileNet which is a more compact than ResNet-18. Table 6 shows the comparison results on MobileNet. AMC [10] is DRL-based static pruning and NetAdapt [34] is heuristics-based static pruning. 75% MobileNet [12] is a variant of MobileNet with width multiplier 75% and input size of 224. Table 6 shows that our method outperforms the comparison methods on both top-1 and top-5 accuracy.

| Method | Baseline top-1 acc. | Top-1 acc. | $\Delta$ top-1 acc. | Baseline top-5 acc. | Top-5 acc. | $\Delta$ top-5 acc. | Speed-up | #Params |
|---|---|---|---|---|---|---|---|---|
| DCP [38] | 69.64 | 67.35 | -2.29 | 88.98 | **88.86** | **-0.12** | 1.71× | 0.71× |
| FPGM [9] | 70.28 | 68.41 | -1.87 | 89.63 | 88.48 | -1.15 | 1.71× | 0.72× |
| Dynamic Sparse Graph [21] | 69.48 | 64.8 | -4.68 | - | - | - | 1.4 × | - |
| CGNN [14] | 69.02 | 67.95 | -1.07 | 88.84 | 88.21 | -0.63 | 1.63× | - |
| FBS [6] | 70.71 | 68.17 | -2.54 | 89.68 | 88.22 | -1.46 | 1.98× | 1.12× |
| AMC [10] | 69.76 | 66.63 | -3.13 | 89.08 | 87.20 | -1.88 | 2.00× | 0.76× |
| Ours ($R_r = 0.5$) | 69.76 | **68.73** | **-1.03** | 89.08 | 88.65 | -0.43 | 1.94× | 0.81× |

Table 5: Comparison with the state-of-the-art channel pruning with ResNet-18 on ImageNet. Speed-up is calculated on MACs.

| Method | Baseline top-1 acc. | Top-1 acc. | $\Delta$ top-1 acc. | Baseline top-5 acc. | Top-5 acc. | $\Delta$ top-5 acc. | Speed-up |
|---|---|---|---|---|---|---|---|
| AMC [10] | 70.9 | 70.5 | -0.4 | 89.5 | 89.3 | -0.2 | 2.00× |
| NetAdapt [34] | - | 69.1 | - | - | - | - | 2.00× |
| 75% MobileNet [12] | 70.9 | 68.4 | -2.5 | 89.9 | 88.2 | -1.7 | 1.79× |
| Ours ($R_r = 0.5$) | 70.9 | **70.6** | **-0.3** | 89.5 | **89.6** | **+0.1** | 2.00× |

Table 6: Comparison with the state-of-the-art channel pruning with MobileNet on ImageNet. Speed-up is calculated on MACs.

# 5 Conclusion

In this paper, we present a deep reinforcement learning based framework for deep neural network channel pruning in both runtime and static sheme. Specially, channels are pruned according to input feature as runtime pruning, and based on entire training dataset as static pruning, with 2 reinforcement agents to determine the corresponding sparsity. Our method combines the merits of runtime and static pruning, and provides trade-off between storage and dynamic flexibility. Extensive experiments demonstrate the effectiveness of our proposed method.

# Acknowledgement

This work is supported by NTU Nanyang Assistant Professorship (NAP) grant M4081532.020, and Singapore MOE AcRF Tier-2 grant MOE2016-T2-2-060 (S).

# Broader Impact

This work is basic research on neural networks compression. We believe this is not applicable to our work.

## Footnotes

[1] https://github.com/deep-fry/mayo

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
