[Reviews · NeurIPS 2020]

Review 1

Summary and Contributions: The author proposes a deep reinforcement learning (DRL) based framework to perform runtime channel pruning on convolutional neural networks. This framework mainly contains three parts: 1) the network that learns two types of channel importance: static channel importance and running time importance; 2) two DRL agents which are producing sparsity ratios during runtime and static pruning procedure. 3) a trade-off pruner to balance the runtime and static pruning results. Experimental results on two benchmark datasets prove that this framework is able to provide a tradeoff between dynamic flexibility and storage efficiency in runtime channel pruning.

Strengths: From the perspective of storage efficiency and dynamic flexibility, the author proposes a pruning method based on reinforcement learning, which combines runtime pruning with static pruning. The author verifies that each part of the network structure has a certain effect through comparative experiments. The research area of the author is related to the theme of NeurIPS community.

Weaknesses: The paper is not innovative enough. What’s more, the experiment part of the paper is not very complete. The number of iterations required to achieve the optimal results is an important indicator, but the paper doesn’t reflect it. I think that hyperparameters have a great impact on the final experimental results. The author did not give a clear explanation of how the hyperparameters involved in the paper should be determined and did not use experiments to illustrate how the hyperparameters are determined. The experimental results show that the method proposed in this paper does not achieve the optimal results in some metrics, and the robustness of the method in this paper is yet to be discussed.

Correctness: Yes

Clarity: Yes

Relation to Prior Work: In the introduction part, the author points out the differences between the method of this paper and the existing methods. In the experimental part, the method proposed in this paper is compared with the existing methods in terms of accuracy, speed-up, number of parameters, and inference time.

Reproducibility: No

Additional Feedback:


Review 2

Summary and Contributions: This paper presents a new channel pruning method by combining the idea of dynamic pruning and static pruning, which can simultaneously reduce storage and runtime. Experiments on CIFAR and ImageNet show the effectiveness of the method.

Strengths: - The idea of combining static and dynamic pruning method is interesting. - The proposed method can outperform both static and dynamic methods.

Weaknesses: - Although combining the static and dynamic pruning method is interesting, the proposed method looks like a incremental work on previous dynamic pruning method, which indeed limits the contribution of this work. - DRL methods usually introduce considerable extra cost. There is no analysis on the cost of the pruning process. - DRL-based methods usually are harder to implement. Since code is not provided, I have concerns on the reproducibility of this paper.

Correctness: I think the claims and method is correct.

Clarity: This paper overall is well organized and easy to read.

Relation to Prior Work: I think the differences from previous work are sufficiently discussed in Introduction and Related Work.

Reproducibility: Yes

Additional Feedback: Overall, I think the proposed idea is interesting and some promising results are achieved. However, I still have some concerns on the learning cost and implementation. As its current state, I would rate this paper as borderline and wait for further discussions. ----- Post rebuttal: The authors' feedback addressed my concerns on the extra cost. After reading other reviews, I think this paper does have some contribution on dynamic pruning methods. Therefore, I raise my score to 6. The authors should provide the code if the paper is accepted.


Review 3

Summary and Contributions: This paper proposed a reinforcement learning based approach to dynamically prune CNN channels in testing time. The agent has two parts: static and dynamic, hence in addition to reduce MACs, the storage footprint is also reduced in some cases. The rebuttal addressed most of my concerns and I slightly raised my score post-rebuttal.

Strengths: +++ The idea of combining static and dynamic pruning in a RL framework is interesting and novel. +++ The abalation studies are pretty good. +++ The experimental resutls are good.

Weaknesses: --- There are one technical error. The Ref [22] is not a dynamic pruning method as claimed in this paper. Ref [22] (Pattern recognition journal, not a arXiv preprint now) had a section devoted to explain how they achieved static pruning. It is, however, approriate to say that the approach in Ref [22] has inspired or been adopted by some dynamic pruning approach. -- Some key information is missing. For example, in tables 1 and 2 and subsequent figures, how are "sparsity" measured? What is the equation that defines this term? -- The empirical comparisons needs some improvements. 1. Wall clock timing is needed. MACs is in fact a bad indicator for dynamic pruning. It surely will slow down (w.r.t. to MAC reduction) when GPU is used. However, I expect the CPU speedup will be somehow closer to MACs reduction. 2. One important result is missing: the comparison with only static pruning using existing method. For example, in table 3, it will be FPGM (or AutoPruner from Ref [22]). For example, on GPU static pruning will have significant speed advantage, which is missing in Table 4, 5, and speed (wall clock) comparison + model size comparison are all missing there.

Correctness: I think they are correct. There are some issues with the experiments, but can be repaired.

Clarity: Mostly yes.

Relation to Prior Work: Mostly yes -- I have pointed out one error in the box above.

Reproducibility: Yes

Additional Feedback:


Review 4

Summary and Contributions: (1) Proposed to prune CNN channels by taking both runtime and static information of the environment into consideration, where runtime information endows pruning with flexibility based on different input instances and static information reduces the number of parameters in deployment, leading to storage reduction. (2) Proposed to use Deep Reinforcement Learning (DRL) to determine sparsity ratios for pruning, which is different from the previous approaches that manually set sparsity ratios. (3) Experiments demonstrate the effectiveness of the proposed method.

Strengths: (1) Soundness of the claims. The proposed DRL method for CNN pruning is well motivated and are theoretically sound. Experiments validate the effectiveness of the method. (2) Significance and novelty. Using DRL to combine the merits of runtime and static pruning is novel, which is also significant as it provides trade-off between dynamic flexibility and memory efficiency. (3) The work focuses on the optimization of neural network architectures, thus is sufficiently relevant to the NeurIPS community.

Weaknesses: Some details are missing, making the method less reproducible, e.g.: (1) In Line 113, M and u are the output of g(u_r; u_s; a^r_t; a^s_t). However, in Sec. 3.2 where the trade-off pruner g(·) is defined, a^r_t and a^s_t are missing. (2) Some implementation details (such as the RNN hyper-parameters) are missing.

Correctness: Yes.

Clarity: The paper is well-structured and easy to follow. There're some minor issues: (1) All variables in Fig. 1 should be represented by math fonts rather than plain text. (2) All vectors variables should be represented consistently (e.g. with bold lowercase), e.g., the decision mask should be m rather than M.

Relation to Prior Work: Yes.

Reproducibility: No

Additional Feedback: As the DRL method doesn't take the channel importance (u_r and u_s) as its action variables (only takes sparsity ratios a^r_t and a^s_t), u_r and u_s cannot get feedback from the reward of the computation/parameters budget, leading to suboptimal solutions. Though taking u_r and u_s as actions requires their sizes to be fixed, they can be then resized to match the input channel number C_in (by down-sampling or interpolation). ==== AFTER REBUTTAL ==== I've read all the reviewing details. The author has addressed my concerns about the reproducibility, but I still feel that the model design lacks optimality (from my comment "8. Additional feedback"). Thus, I will keep my score.

[Author Response · NeurIPS 2020]

**Reviewer #1: Q1:** The number of iterations ... **A1:** We did not present values of all the hyper-parameters in submission due to the limit of space. Hyper-parameters are described as follows. The batch size for training is 128. For CIFAR-10, we train ($N_1 = 1560$) batches (4 epochs) for DRL agents (corresponding to line 12-15 in Algorithm 1) and ($N_2 = 780$) batches (2 epochs) for $h(\cdot)$, $v$, $f(\cdot)$ and CNN (line 16-19 in Algorithm 1) at each iteration. The total number of iterations is 40 (for line 1-19 in Algorithm 1). For ImageNet, at each iteration of training, $N_1 = 1200$ and $N_2 = 600$ except last iteration. The total number of iterations is 64. At last iteration, $N_2 = 200,000$ for finetuning the CNN. The learning rate for DRL agent is $10^{-6}$. Learning rate for CNN is $10^{-3}$ in CIFAR-10 and $10^{-4}$ in ImageNet.

| DRL lr | CNN lr | Acc. |
|---|---|---|
| $10^{-6}$ | $10^{-3}$ | **91.23** |
| $10^{-6}$ | $10^{-2}$ | 10.00 |
| $10^{-6}$ | $10^{-4}$ | 90.67 |
| $10^{-5}$ | $10^{-3}$ | 89.95 |
| $10^{-7}$ | $10^{-3}$ | 90.79 |

We show experiment results on CIFAR-10 with various learning rate setting in table on the right side. The sparsity is 0.5 and $R_r = 0.5$. It shows that our learning rate setting is optimal.

**Q2:** The experimental results **A2:** We show the training curve of our method. We train on CIFAR-10 with sparsity 0.5 and $R_r = 0.5$. The y-aix is test accuracy and x-aix is the number of epochs. This figure shows that our method reaches optimal and robust during training.

**Reviewer #2: Q1:** Although combining ... **A1:** Our method combines static and runtime pruning under a DRL-based framework which determines sparsity of layers and addresses the storage efficiency problem of runtime pruning. Our method takes advantages of both static and runtime pruning to provide trade-off with better accuracy and lower storage.

**Q2:** DRL methods usually ... **A2:** For training cost of DRL, kindly refer to **Reviewer #1 A1**. The extra cost of DRL at inference is tiny comparing to convolutional layers. The extra cost of each layer includes the computation of layer-dependent encoder (described in paragraph **State** in Sec. 3.3 ) and runtime DRL agent. The layer-dependent encoder is a fully-connected layer where input size is input channel of this convolutional layer and output size is 128. The runtime DRL agent network consists of one layer of RNN with hidden state size of 128 and one layer fully-connected Actor network with output size of 1 (the mean of Gaussian policy). The MACs of layer-dependent encoder and runtime DRL agent is around $0.3\%$ of pre-trained CNN.

**Q3:** DRL-based methods usually ... **A3:** We are preparing the code and will open-source after paper is accepted.

**Reviewer #3: Q1:** There are one... **A1:** The training process of Ref[22] is dynamic pruning but in inference it is static pruning. We will revise the related work to claim Ref[22] as static pruning and also update bibtex of Ref[22].

**Q2:** The "sparsity" of Table 1 and 2 means the ratio of preserved output channels after pruning at every layer. This "sparsity" setting is mainly for FBS[22] because FBS uses a same sparsity value for all convolutional layers. However, our proposed method predicts layer-specific sparsity ratios by DRL agents. Therefore, for our method in Table 1 and 2, we calculate the computation and storage budget constraints according to the "sparsity" value (which is 0.5 in Table 1 and 0.7 in Table 2), then our approach learns layer-specific sparsity ratio according to the constraints by DRL method.

**Q3:** Wall clock ... **A3:** The "Inference Time" in Sec. 4 Table 1 is wall clock of running on GPU. We show the CPU wall clock and static pruning method FPGM in following table. We also show experiment result of FPGM which is static pruning approach in this table. Since it is static pruning method, it has smaller wall clock but lower accuracy compared to runtime pruning approach FBS and our method.

| Method | Acc. | $\Delta acc.$ | GPU Time | CPU Time |
|---|---|---|---|---|
| FBS | 89.88 | -1.49 | 10.9 ms | 172.0 ms |
| RNP | 84.93 | -7.14 | 11.1 ms | 175.3 ms |
| FPGM | 89.8 | -2.27 | **5.0 ms** | **48.2 ms** |
| ours ($R_r = 1$) | 91.425 | **-0.645** | 11.2 ms | 178.3 ms |
| ours ($R_r = 0.5$) | 91.228 | **-0.842** | 9.8 ms | 110.7 |

**Q4:** One important ... **A4:** For the comparison of FPGM, please refer to **A3**. We report model size (number of parameters stored) of methods pruning ResNet-18 on ImageNet in following table. Static method can reduce more parameters (storage consumption) while runtime method FBS increases the number of parameters. Our method can reduce more storage compared to runtime method. The wall clock comparison of Table 4 and 5 in Sec. 4 is unavailable, for the method doesn't report actual time nor release codes for re-run. Although some methods released codes, it is not fair to compare wall clock because their implementation is not optimized for latency.

| Method | $\Delta$ top-1 acc. | $\Delta$ top-5 acc. | Speed-up | #Params |
|---|---|---|---|---|
| DCP | -2.29 | **-0.12** | 1.71× | 0.71× |
| FPGM | -1.87 | -1.15 | 1.71× | 0.72× |
| Dynamic Sparse Graph | -4.68 | - | 1.4× | - |
| CGNN | -1.07 | -0.63 | 1.63× | - |
| FBS | -2.54 | -1.46 | 1.98× | 1.12× |
| AMC | -3.13 | -1.88 | 2.00× | 0.76× |
| Ours ($R_r = 0.5$) | **-1.03** | -0.43 | 1.94× | 0.81× |

**Reviewer #4: Q1:** In Line 113, ... **A1:** We will carefully revise the definition of trade-off pruner in our next version of paper. Here let me clarify the trade-off pruner: In Sec. 3.2, we don't mentioned $a_t^r$ and $a_t^s$ because we use the mask $\mathbf{M}_r$ and $\mathbf{M}_s$ to represent the pruning results which are computed according to $a_t^r$ and $a_t^s$. $\mathbf{M}_r$ is computed by $\mathbf{u}_r$ and $a_t^r$ as mentioned in line 134-139 of Sec. 3.1 and $\mathbf{M}_s$ is computed by $\mathbf{u}_s$ and $a_t^s$ as mentioned in line 163-166 of Sec. 3.1.

**Q2:** Some implementation ... **A2:** We use 1-layer RNN with hidden state size of 128. For more implementation details, please refer to **Reviewer #1 A1** and **Reviewer #2 A2** line 19 to 22.

**Q3:** All variables in Fig. 1 should ... **A3:** Thanks for your suggestion. We will update the font in Fig.1.

**Q4:** All vectors variables should ... **A4:** We will change decision mask to $\mathbf{m}$ and will carefully revise the matrix/vector variables in math equations.

[Meta-Review · NeurIPS 2020]

This is an interesting paper that combines static and dynamic pruning of CNN channels, adding an RL agent into the loop is still able to provide an overall speed up in inference. The reviewers were concerned the paper did not describe how the hyperparameters were chosen and that the choice of action space was not optimal, thus the authors are encouraged to further clarify this in subsequent versions of the paper.